# Molecular evidence of pathogens and endosymbionts in the black horse fly *Osca lata* (Diptera: Tabanidae) in Southern Chile

**Christian R. González**[1]*, Carolina Reyes[2], Andrés Castillo[2], Lara Valderrama[2], Lorena Llanos[2], Jorge Fernández[2], Gillian Eastwood[3]*, Beatriz Cancino-Faure[4]

1 Instituto de Entomología, Facultad de Ciencias Básicas, Universidad Metropolitana de Ciencias de la Educación, Santiago, Chile, 2 Sección Entomología y Genética Vectores, Sección Genética de Agentes Infecciosos, Subdepartamento de Genómica y Genética Molecular, Instituto de Salud Pública de Chile, Santiago, Chile, 3 Department of Entomology, College of Agriculture and Life Sciences, Virginia Polytechnic Institute and State University, Blacksburg, Virginia, United States of America, 4 Laboratorio de Microbiología y Parasitología, Departamento de Ciencias Preclínicas, Facultad de Medicina, Universidad Católica del Maule, Talca, Chile

* christian.gonzalez@umce.cl (CRG); geastwood@vt.edu (GE)

**Data Availability Statement:** The data that support the findings of this study are publicly available from Genbank (from 3rd July 2024) with the following

## Abstract

Little is known about the role of horse flies in potential pathogen transmission in Chile. This study provides evidence of the molecular detection of microorganisms in southern Chile. In the present study, adult *Osca lata* horse flies were trapped from Punucapa (39˚45'06"S/73˚16'08"W, Región de Los Ríos) and Puyehue (40˚39'10"S/72˚10'57"W, Región de Los Lagos), Chile. Among the 95 samples analyzed by PCR using specific primers, microorganisms were detected in 23.2% (n = 22) of the samples. *Rickettsia* spp. DNA was detected in 15.8% (n = 15) of the samples, *Trypanosomatidae* DNA in 5.3% (n = 5) of the samples, and filarial DNA in 2.1% (n = 2) of the samples. This study found that horse flies in the region are capable of carrying a variety of both parasites and endosymbionts. Further research is needed to understand the specific impact of horse flies as mechanical or biological vectors and develop effective control measures to prevent the spread of any microorganisms associated with disease.

## Author summary

Horse flies are well-recognized mechanical vectors of pathogens in humans and domestic animals. Our study on the black horse fly, *Osca lata*, an abundant species in south-central Chile, sought to explore whether this hematophagous species could act as a vector of microorganisms that can affect domestic animal, and even human, health. We used molecular techniques to detect evidence of three types of microorganisms in samples analyzed from southern Chile. These results indicate that this fly species has the potential to harbor parasitic agents. Further studies are necessary to understand the specific role of horse flies as mechanical vectors and their potential impact on disease transmission in different regions and animal populations.

identifiers/ accession numbers: PP949209
PP949196 PP949197 PP949198 PP949199
PP949200 PP949201 PP949202 PP949203
PP949204 PP949206 PP949207 PP949208
PP326074/PP949195 PP343119 PP343120
PP343121 PP326075 The data that supports the
findings of this study are publicly available at the
Open Science Framework repository. Datasets can
be found at https://osf.io/y4m5k.

**Funding:** This work was supported partially funded
via the Dirección de Investigación de la Universidad
Metropolitana de Ciencias de la Educación
(DIUMCE) (06-2021-PGI) (to CRG). The funders
had no role in study design, data collection and
analysis, decision to publish, or preparation of the
manuscript.

**Competing interests:** The authors have declared
that no competing interests exist.

## Introduction

Hematophagy is a highly specialized feeding habit that has evolved in various lineages of both invertebrates and vertebrates. It is estimated that this habit emerged independently at least 20 times during the evolution of Arthropoda, approximately 165–145 million years ago, in the Jurassic and Cretaceous periods [1]. In invertebrates, hematophagy is observed in four groups of Chelicerata, the most significant being of the Order Ixodida (ticks), acting as a vector for arboviruses, bacteria, and protozoa that can affect the health of humans and various animal groups [2]. Within Hexapoda, five orders exhibit this feeding mode, although it is more commonly observed in Diptera, where it occurs in 16 families encompassing over 9,000 hematophagous species [1]. Indeed, the hematophagous feeding strategies observed in various lineages of invertebrates, particularly within the Arthropoda, have significant implications for both humans and animals. By feeding on blood, these hematophagous invertebrates have the potential to transmit pathogens and cause serious diseases such as malaria, dengue, or leishmaniasis [1].

Horse flies (Diptera: Tabanidae) are hematophagous flies that, in different parts of the world, cause discomfort to humans and domestic animals by flying insistently around their potential prey and trying to land on them to bite with their specialized mouthparts. Horse flies are typically considered to be mechanical vectors rather than biological vectors [3]. Mechanical vectors are organisms that can physically transmit pathogens without being an essential part of their life cycle, replication or development. Horse flies could serve as potential mechanical vectors by carrying pathogenic agents on their mouthparts or bodies directly from an infected host or indirectly via contaminated substrate, to a susceptible host [4]. Horse flies are elsewhere known to be mechanical vectors for pathogens in different regions of the world [5]. Although they are primarily considered nuisance pests owing to their painful bites, their feeding habits can contribute to the transmission of diseases [3,6–9].

Horse flies feed on the blood of various animals including mammals. During their blood-feeding activities, they come into contact with microorganisms (pathogenic or otherwise), such as bacteria or parasites, present in the blood of infected animals. If they subsequently feed on another animal, they may transfer these parasites, potentially leading to disease transmission. The blood-feeding style seems unsuitable for the efficient transmission of arthropod-borne viruses (arboviruses) to animals; however, Kobayashi et al. [10] do report the presence of an insect-specific *flavivirus* in the horse fly *Tabanus rufidens* Bigot, 1887. This finding opens new areas for the exploration of the real role of tabanids in the potential transmission of arboviruses.

It is important to note that the role of horse flies as mechanical vectors in disease transmission may vary depending on the specific pathogen involved, environmental conditions, and host factors. Although they have been implicated in the mechanical transmission of certain pathogens [3,5], several of which are discussed below, their overall significance as disease vectors may be lower than that of biological vectors, such as mosquitoes and ticks, which are involved in the life cycle of the pathogens they transmit. Nevertheless, at least 33 pathogens have been associated with and transmitted by horse flies either biologically or, more frequently, mechanically. Of these 33 agents, 16 have a worldwide occurrence, seven have been reported for Africa, six for Europe, five for North America and four for Asia and South America [3]. Tabanids are biological vectors of filarial nematodes (i.e., *Elaeophora schneideri*, *Loa loa*, *Dirofilaria roemeri*, and *D. repens*), and protozoa (e.g., *Haemoproteus metchnikovi* and *Trypanosoma theileri*). However, they are also important in the mechanical transmission of pathogens and have been associated with the transmission of six viruses (e.g., Equine infectious anemiavirus, Bovine leukosis virus, Vesicular stomatitis virus), eleven bacteria (e.g., *Francisella tularensis*, *Bacillus anthracis*, *Clostridium chauvoei*, and *Listeria monocytogenes*), and eight

protozoa (e.g., *Besnoitia besnoiti*, *Trypanosoma evansi*, *T. vivax*, and *T. brucei*) [3]. Probably, the growing interest in this group of Dipteran insects as vectors will increase the knowledge and reports in different regions of the planet, as has happened in South America.

Several studies in South America have explored the presence of pathogens in different horse flies species using molecular techniques. Fermino et al. [11] found that *Trypanosoma* spp. are associated with two species of horse flies, *Phaeotabanus fervens* Linnaeus 1758 and *T. occidentalis* Linnaeus 1758. Rodrigues et al. [12] detected *Anaplasma marginale* in three *Tabanus* species and one *Dasybasis* species in Uruguay. Rodrigues et al. [13] detected *T. kaiowa* in *T. triangulum* in Rio Grande do Sul, Brazil. Ramos et al. [14] recorded DNA of *T. evansi*, the etiological agent of surra, in *Dichelacera alcicornis* and *D. januarii*, in South America. There have however, been no previous reports of pathogen detection within tabanids in Chile, nor understanding of its associations with non-pathogenic microorganisms or endosymbionts.

*Osca lata* is a hematophagous and abundant tabanid species in south-central Chile (Región Metropolitana to Región de Los Lagos) and Argentina (in the provinces of Neuquén and Río Negro). Once the adults emerge (end of December), and there were no heavy or prolonged rains, their abundance can be observed to increase rapidly until the adults decrease and disappear around the end of January. The economic impact that this species causes in the areas where it is most abundant (Araucanía Region to Los Lagos Region) has not been evaluated, but the aggressive nature of the females in search of blood must affect the cattle production and tourism that takes place during January (*O. lata* peak abundance) in the lake area of southern Chile.

Studies focusing on horse flies as potential vectors of pathogens play a crucial role in understanding the potential role of horse flies as vectors in the region, and can contribute to our knowledge of disease ecology, zoonotic risks, and the overall understanding of vector-borne diseases in South America.

Here we use molecular techniques, aiming to investigate the presence of microorganisms in the black horse fly, *Osca lata*, in Chile. We reveal novel findings of parasite association in this tabanid species, which could serve as a potential mechanical vector in the region for potentially pathogenic microorganisms.

## Material & methods

### Ethics statement

Ethical approvals for this sampling in this study were obtained from the Ethics Committee of Universidad de Santiago, Chile (approval number 038/2021).

### Study site

The research was conducted in Punucapa (Región de Los Ríos) and the peripheral area of the Puyehue National Park (Región de Los Lagos), Chile (Fig 1). The climate of the whole region is rainy temperate, with abundant rainfall throughout the year and an average annual rainfall of more than 3,000 mm.

### Sample collection

Tabanids were collected for 3 h in the morning (10:00 am to 13:00 pm) and 3 h in the afternoon (15:00 p.m. to 18:00 p.m.) on January 2021 and January 2022 from two localities: 1) Punucapa (39°45'06"S/73°16'08"W) and 2) Puyehue (40°39'10"S/72°10'57"W), using an entomological net. After collection, each fly was placed in an individual plastic bottle and preserved in 80% ethanol for further morphological and molecular processing. Taxonomic identification was performed according to dichotomous keys [15].

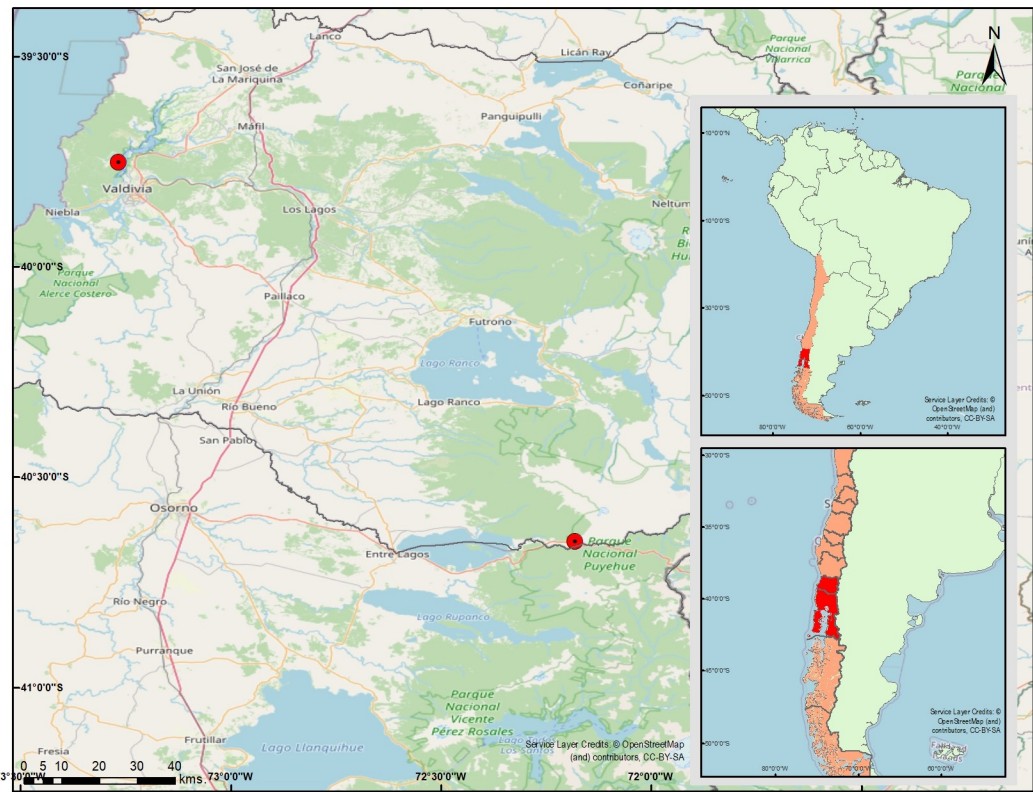

**Fig 1. Map of southern Chile showing tabanid sampling sites.** (Base layers used in the map correspond to the administrative boundaries published open access by the Secretariat of Regional and Administrative Development of IDE Chile—Ministerio de Bienes Nacionales).

### Tabanid dissections

Tabanids were rinsed in sterile phosphate buffered saline (PBS; 10 mM $Na_2HPO_4$, 150 mM NaCl, pH 7.4). Using sterile forceps and dissecting needles, tabanid tissues were isolated and thoroughly rinsed in PBS. The proboscis and its components, the proventriculus, midgut, hindgut, and rectum, were dissected for downstream molecular analysis. Dissected tissues from each fly were placed into separate 1.5 mL microfuge tubes in preparation for DNA extraction.

### Sample preparation, DNA extraction and quantification

The mouthparts and intestinal contents of each specimen were used for DNA extraction using the QIAamp DNA Mini Kit (QIAGEN, Germantown MD, USA) according to the manufacturer's instructions. Dried tabanid samples were first pulverized (ground) with a sterile pestle in a 1.5 mL tube. All samples were eluted in 100 μL of ultrapure water and quantified using a Qubit dsDNA Quantification Broad-Range assay kit (Invitrogen, USA).

### Pathogen detection

Primers from other authors, as well as those specially designed for this study, are listed in Table 1. All PCR amplifications were performed in a final volume of 50 μL, containing 25 μL of SapphireAmp Fast PCR Master Mix (Takara Bio, USA), 2 μL (20 pmol) of primer, 5 μL of DNA template, and 18 μL of nuclease-free water. For *Rickettsia* detection, a two-step PCR and

**Table 1. List of primers used for pathogen detection.**

| | Sequence | Gene | Product size | Reference |
|---|---|---|---|---|
| *Rickettsia* First PCR | GltA-F; 5'—TCCTATGGCTATTATGCTTG—3'<br>GltA-R; 5'- CCTACTGTTCTTGCTGTGG—3' | *gltA* | 789 bp | [16] |
| *Rickettsia* semi-nested PCR | GltA-R2; 5'-ACCGTGAACATTTGCGACGGTAT -3' | *gltA* | 760 bp | this study |
| *Trypanosomatidae* | F 720; 5'-GTTAAAGGGTTCGTAGTTGAA-3'<br>R1495; 5'-GACTACAATGGTCTCTAATCA-3' | *18S* | 600 bp | [17] |
| *Filarioidea* | Fwd.18S.631; 5'-TCGTCATTGCTGCGGTTAAA-3'<br>Rwd.18S.1825r; 5'-GGTTCAAGCCACTGCGATTAA-3' | *18S* | 1151 bp | [18] |

semi-nested PCR strategy was employed to increase the specificity in *gltA* gene amplification. The initial PCR was performed at 95˚C for 5 min, followed by 40 cycles at 95˚C for 30 s, 48˚C for 30 s, 72˚C for 30 s, and a final extension for 7 min at 72˚C. The subsequent semi-nested PCR, using a product of the first PCR, was performed at 95˚C for 5 min, followed by 40 cycles of 95˚C for 30 s, 55˚C for 30 s, 72˚C for 30 s, and a final extension for 7 min at 72˚C. For *Trypanosomatidae* detection, conventional PCR was performed as follows: 95˚C for 5 min, followed by 40 cycles of 95˚C for 30 s, 58˚C for 30 s, 72˚C for 30 s, and a final extension for 7 min at 72˚C. For detection of Filarioidea species, cycling conditions were 95˚C for 5 min, followed by 40 cycles of 95˚C for 30 s, 54˚C for 30 s, 72˚C for 30 s, and a final extension for 7 min at 72˚C. All PCR products were separated by electrophoresis on a 2% agarose gel stained with SYBR Safe DNA (Invitrogen), with a ~760 bp band expected for *Rickettsia*, ~600 bp for *Trypanosomatidae*, and ~1151 bp product for Filarioidea.

## DNA sequencing

To identify the microorganisms found by PCR in the studied samples, we sequenced and analyzed specific genes depending on the specific species of interest and constructed a phylogenetic tree. Purified PCR products were sequenced in both directions using the BigDye Terminator v3.1 cycle sequencing Kit (Applied Biosystems, USA) on an ABI 3500 Genetic Analyzer (Applied Biosystems, USA).

## Phylogenetic analysis

Nucleotide sequences in FASTA format obtained from electropherograms were analyzed using the online version of the BLASTn (NCBI, USA) tool against the GenBank database. The sequences with the highest BLAST scores were recovered from the NCBI database for multiple sequence alignment performed using MAFFT v7.4 alignment tool. Best alignments were used as input for the maximum likelihood phylogenetic reconstruction using the IQ-TREE software V2.0.3 [19] with a bootstrap of 1000 replicates.

## Results

A total of 95 field-caught female specimens of *O. lata* were collected and screened: 14 in the locality of Punucapa (Región de Los Ríos) and 81 in Puyehue (Región de Los Lagos). Of these, 22 specimens (eight in 2021 and 14 in 2022) tested positive for one or more of the detected microorganisms.

## Data description and microorganism identification

Of 95 samples analyzed by PCR, microorganisms were detected in 23.2% (n = 22) of the samples. Specifically, *Rickettsia* spp. DNA was present in 15.8% (n = 15) samples,

**Table 2. Co-infection detected in specimens of *Osca lata*.**

| Year | Place | Number specimens | Microorganism detected using BLAST |
|------|-------|------------------|-------------------------------------|
| 2021 | Puyehue | 1 | Filaria/*Rickettsia* |
| 2022 | Puyehue | 1 | *Trypanosomatidae/Rickettsia* |

*Trypanosomatidae* DNA in 5.3% (n = 5), and filaria DNA in 2.1% (n = 2). Out of the 95 samples analyzed in our study, two samples (2.1%) exhibited co-infection with two distinct microorganisms. Specifically, one specimen was co-infected with Filarioidea and *Rickettsia*, while another specimen showed co-infection with *Trypanosomatidae* and *Rickettsia* (Table 2). DNA from this microorganism was detected only in specimens collected from Puyehue.

## Phylogenetic analysis

PCR amplicons were sequenced and analyzed according to the best matches in the NCBI nucleotide database using BLAST. The tree topology of the *Rickettsia* genus was determined using sequences of the *gltA* gene (Fig 2). Sequences included in the phylogenetic reconstruction are representative of the positive samples obtained by semi-nested PCR. Samples PP949195, PP949196, PP949197, PP949198, PP949199, PP949200, PP949201, PP949202, PP949203, PP949204, and PP949205 were grouped into a clade separate from that of PP949206, PP949207, PP949208 and PP949209. Although they are grouped in different clades, they are highly similar in their nucleotide identities (96.1% to 99.3%), and they are grouped closest to unidentified *Rickettsia* endosymbiotic of *Bemisia tabaci* (Hemiptera: Aleyrodidae), reported in populations of invasive whitefly, with 97.7% to 98.3% identity [20].

For *Trypanosomatidae* identification, phylogenetic reconstruction was performed based on nucleotide sequences with the highest scores in the BLAST search. The topology of the

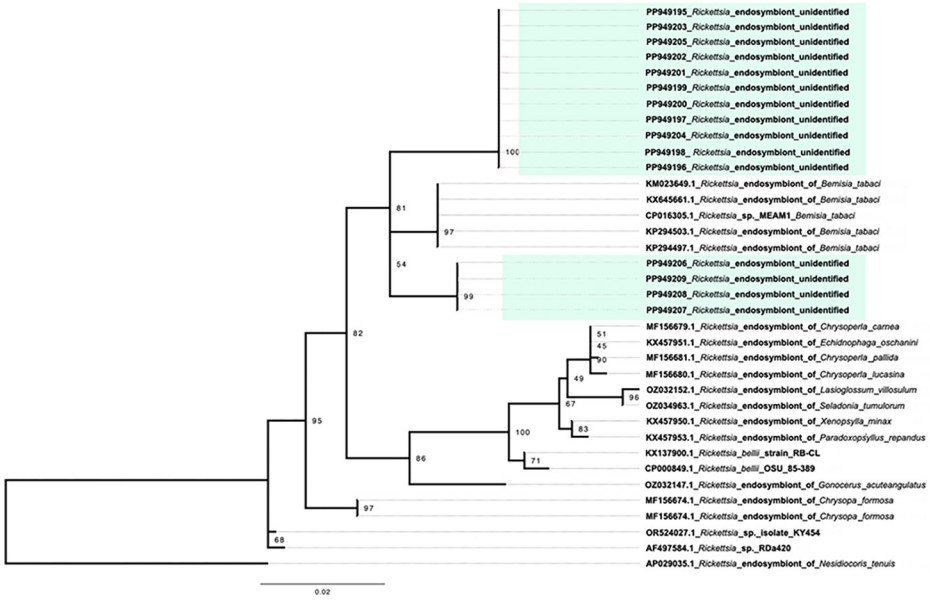

**Fig 2. Phylogenetic analysis of *Rickettsia* spp. bacteria detected in *Osca lata*.** The evolutionary history based on the *gltA* gene was inferred using the maximum likelihood method with HKY+F+R2 as the best-fit model. The tree was drawn to scale, and branch lengths indicate the number of nucleotide substitutions per site. Chilean samples from this study are highlighted in green, and the reference sequences are labeled with the NCBI Nucleotide entry. A total of 740 nucleotide positions were aligned for phylogenetic inference.

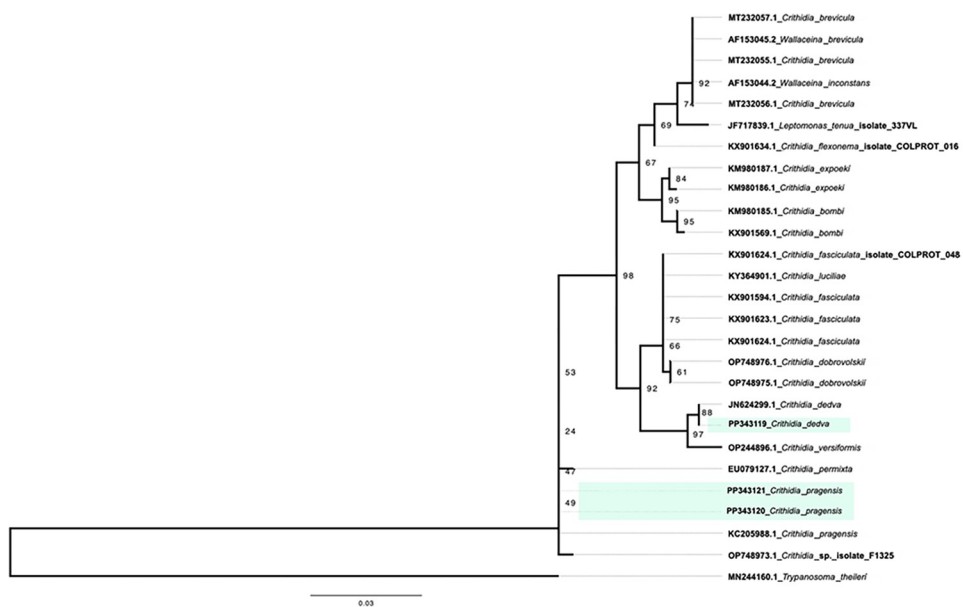

**Fig 3. Phylogenetic analysis of *Trypanosomatidae* detected in *Osca lata*.** Evolutionary history based on the *18S* rRNA gene was inferred using the maximum likelihood method with TNe+G4 as the best-fit model. The tree was drawn to scale, and branch lengths indicate the number of nucleotide substitutions per site. Chilean samples are highlighted in green, and the reference sequences are labeled with the NCBI Nucleotide entry. A total of 522 nucleotide positions were aligned for phylogenetic inference.

phylogenetic tree (Fig 3) locates the sample PP343119 constituting a single cluster with the protozoan parasite *Crithidia dedva* with 100% nucleotide sequence identity. Additionally, samples PP343120 and PP343121 were grouped together in a separate cluster, where the closest match was *C. pragensis*, with 99.8% sequence identity.

Filarioidea identification was performed by 18S rRNA gene sequencing. Phylogenetic reconstruction (Fig 4) showed that the two identified samples, PP326074 and PP326075, shared the same clade, closely related to *Dirofilaria immitis*, with an identity percentage between 99.5 and 99.6.

## Discussion

Here we describe molecular evidence of microorganisms in *Osca lata*. This is the first study to identify such microorganisms in horse flies in Chile, and in this species. The fly is of concern given that *O. lata* females avidly seek blood from different hosts, including humans. Their nuisance-biting alone causes unquantified losses to two economically important industries in Chile, firstly affecting tourism, an industry which develops in the southern zone of the country in the summer months (December-February), and secondly to productivity of the livestock industry with hematophagy disrupting livestock feeding impacting weight gain and milk production [21,22]. However, whether this species carries microorganisms in the country had not been assessed to date. Here we show the association of *O. lata* with three different groups of parasite (bacterial, protozoa, and filarial worms), with pathogenic *D. immitis* being a known pathogen, and other microorganisms detected being either an endosymbiont, or having a role yet to be determined.

Elsewhere, due to their behavior, biting habits, characteristics of the mouthparts, and high abundance of some species in different regions of the planet, Tabanids have been considered efficient mechanical vectors [23,24].

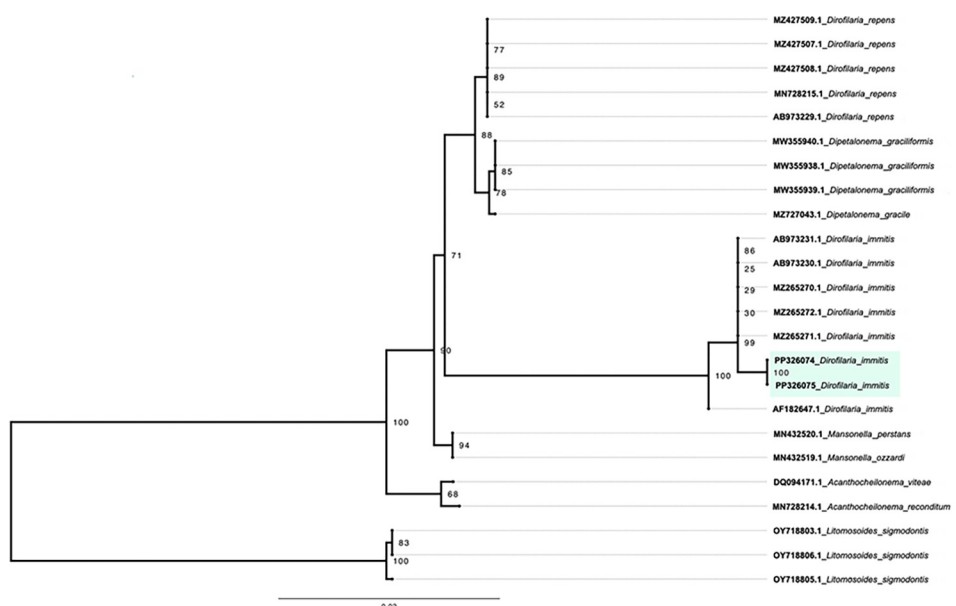

**Fig 4. Phylogenetic analysis of Filarioidea species detected in *Osca lata*.** Evolutionary history based on the *18S* rRNA gene was inferred using the maximum likelihood method with K2P+I as the best-fit model. The tree was drawn to scale, and branch lengths indicate the number of nucleotide substitutions per site. Chilean samples are highlighted in green, and the reference sequences are labeled with the NCBI Nucleotide entry. A total of 1083 nucleotide positions were aligned for phylogenetic inference.

Novel molecular techniques have led to advances in determining the presence of microorganisms in tabanid species worldwide. In Chile, there were no reports on the presence of parasites in different species of tabanids. Molecular data indicated that the highest prevalence of microorganisms in this tabanid species collected in southern Chile was for genus *Rickettsia* (15.8%) followed by *Trypanosomatidae* (5.3%), and filarial worms (2.1%).

In Africa [17,25], Asia [26], Oceania [7], and Europe [27], several tabanid species have been syndicated as vectors of protozoan parasites. In South America, the prevalence of *Trypanosomatidae* species reported in several tabanid species, particularly *T. evansi* and *T. kaiowa*, varies from 40 to 33% [11], for species feeding on caiman species (Alligatoridae: Caimaninae); Rodriguez et al. [13] reported the detection of *T. kaiowa* in 37.5% in *T. triangulum*, extending the pathogen distribution range to southern Brazil (Rio Grande do Sul); and Ramos et al. [14] report the presence of *T. evansi* in two species of *Dichelacera* horse fly. Recently, Chávez-Larrea et al. [28] reported *T. theileri* in bovine in Ecuador suggesting the participation of tabanids. Our results did not detect any *Trypanosoma* species, with two samples close to *C. pragensis* (Fig 3), a recently described species of *Trypanosomatidae* that would be typical of Brachycera Diptera [29], whose host type is *Cordilura albipes* (Diptera: Scathophagidae) and one sample close to *C. dedva*, a parasite also found in *Culex pipiens* [30]. Several species of *Trypanosomatidae* monoxenous have been found in horse flies and *C. rileyi* was described from the hindgut of *T. epistates* from Minnessota (USA) [31]. These *Trypanosomatidae* species belongs to the subfamily Leishmaniinae [32], and the species detected in the analyzed Chilean samples, was close to a monoxenous *Trypanosomatidae C. pragensis* and *C. dedva*, which would have *O. lata* as its only host. However, the presence of these monoxenous *Trypanosomatidae*, recognized pathogens of invertebrates, mainly insects, has been reported in immunocompromised or *Leishmania*-affected individuals [33,34]. Therefore, further surveillance and research is required to confirm the pathogenicity of this monoxenous parasite, which constitutes a potential risk situation for people in the distribution area of *O. lata*.

Different tabanid species have been reported as biological vectors of various filarial nematode species in different zoogeographic regions. The most widely known filarial species, transmitted by several species of *Chrysops* (two main *C. silacea* and *C. dimidiata*), is *Loa loa*, an agent of human loiasis known as the tropical eye worm, which is endemic to Central Africa. Recent studies have demonstrated that excessive mortality is associated with high infection levels [35]. Spratt [36] reported the presence of third-stage larvae of *Dirofilaria roemeri* in two species of Australian distribution: *Dasybasis oculata* and *T. parvicallosus*. According to Spratt [36], the presence of *D. roemeri* has a detrimental effect on the host and tabanids, and the host-seeking ability of these tabanid species is not impaired. Experimental studies have shown that *Dirofilaria* species can develop in numerous arthropods, but only mosquitoes (Diptera: Culicidae) act as biological vectors; during mosquito bite, an infected mosquito, introduces third-stage filarial larvae into definitive host [37]. Grunenwald et al. [38] reports the transmission of *Elaeophora schneideri* (Spirurida: Onchocercidae) to moose (*Alces alces*) in Minnesota (USA) by *Chrysops* spp. deer flies and *Hybomitra* spp. horse flies.

The presence of *D. immitis*, which causes dirofilariasis, in the samples analyzed in our study, suggests that the parasite circulates in the area. However, to determine whether *O. lata* is a competent vector of *D. immitis*, L3 larvae should be found in the tabanids and the DNA of *D. immitis* in the blood-fed tabanids. Studies conducted on dogs and mosquitoes, [39–41] in the central zone of Chile, have failed to demonstrate the presence of this filaria. Larger studies should be undertaken to thoroughly investigate their presence. *D. immitis* has been reported as endemic in Argentina [42,43] and other neighboring countries and here there is only a report of an imported animal originating from an endemic zone [44].

The detection of *D. immitis* in the *O. lata* specimens collected in this study could indicate that the feeding of this species of horse fly could include fox species (Carnivora: Canidae), which carry this pathogen undetected. Studies on feeding patterns of *O. lata* are needed to determine the vector's feeding patterns and the possible risk to people in the range of this species.

*Rickettsia* is an obligate intracellular bacterium best known as the causative agent of human and animal diseases, transmitted by hematophagous arthropods through salivary secretions or feces [45]. It has been estimated that 24% of arthropod species harbor *Rickettsiae*, with many of them serving as endosymbionts [46]. The occurrence of *Rickettsia* spp. in tabanids has not been widely reported in previous studies [47], although species of the family Anaplasmataceae have been frequently reported in tabanids [3,5,47]. Keita et al. [17] mention the presence of three *Rickettsia* species associated with only one species of tabanids, *Atylotus fuscipes*, in West Africa. Several other bacterial species have been reported in the literature as being mechanically transmitted by various species of tabanids [3, 5]; for example, Hornok et al. [47] detected the presence of *An. marginale* in *Tabanus bovinus* and claimed that this tabanid species could be an even more important vector mechanically than the biological vector, hard ticks. This is supported by the facts that painful feeding is usually interrupted by hosts several times and that the important flight capacity of the vector, which can travel several kilometers. Hawkins et al. [48] demonstrated that females carrying *Anaplasma* spp. remain infective for at least two hours after obtaining a partial blood diet from an infected host. In North America, Scoles et al. [49] reported the vector action of *T. fuscicostatus* in transmitting *An. marginale* in Asia. In South America, Rodrigues et al. [12] reported *An. marginale* in Uruguay. Clearly, the presence of *Rickettsia* in tabanids is scarce, and the explanation of its high prevalence in *O. lata* is not yet available, but determining which species it is, we could conjecture about the epidemiological implications of the findings. We cannot rule out that it is a group of *Rickettsia*, such as those in the Torix group, which appear to reside exclusively in invertebrates and protists with no secondary vertebrate host [50]; these were initially detected in groups of aquatic insects

[51], but have recently been mentioned in groups of terrestrial insects such as Cimicidae (Hemiptera), Curculionidae (Coleoptera), and Calliphoridae (Diptera) [50]. The closest matches of our gene sequences with the references from GenBank are related to *Rickettsia* endosymbiotic of *Bemisia tabaci* [20] and endosymbiotic of arthropod like species from genus *Chrysoperla*, *Chrysopa*, *Xenopsylla*, *Echidnophaga*, *Paradoxopsyllus*, *Seladonia*, among others.

For accurate identification of *Rickettsia* species, additional gene analysis assays are necessary to achieve species-level identification. Despite the public NCBI database having nucleotide sequences closely related to our samples, bacterial isolation and strain enrichment may be necessary for better identification, through 16S and/or whole genome sequencing. This approach will help determine whether our findings correspond to a known species or potentially represent a novel species not previously described. According to NCBI database resources, we identified more than 99.5% of nucleotide identity species within the family *Trypanosomatidae* and species belonging to the superfamily Filarioidea with high certainty.

The distribution area of *O. lata* in Chile is extensive, more than 1,400 km, and there are other hematophagous arthropods that could act as reservoirs or competent vectors of microorganisms in the region. Considering mosquito species, no studies have been conducted to detect the presence of viruses or protozoa, although Cancino-Faure *et al*. [41] did not detect the presence of *Dirofilaria* spp. in a study where they analyzed *Ae*. (*Och*.) *albifasciatus*. In ticks, Abarca *et al*. [52] detected the presence of "Candidatus *Rickettsia andeanae*" in *Amblyomma tigrinum* in Angol. Ivanova *et al*. [53] described a new species of *Borrelia*, *B. chilensis*, a member of *B. burgdorferi s.l.* complex, in Valdivia isolated from *Ixodes stilesi* collected in long-tailed rice rats (*Oligoryzomys longicaudatus*). In trombiculid mites, Acosta-Jamett *et al*. [54] reports the presence of rodent-associated chigger mites positive for *Orientia* sp. in southern Chile (Chiloé).

Further epidemiological studies are necessary to better understand the specific role of horse flies as mechanical vectors and their potential impact on disease transmission in different regions and animal populations. Adopting a One Health approach to generate understanding of vector-host interactions (both human and wild or domestic animals), host competency and exposure, would thus be beneficial in defining role of *O. lata* in pathogen transmission, and in designing integrated strategies to mitigate the risk of vector-borne disease. Sampling of domestic and wild animals that may act as reservoirs for the pathogens detected, assessment of host species and geographic ranges, understanding life cycles and developmental stages of both flies and pathogens, and the potential pathogenicity of any newly discovered species, are important tasks that will need to be implemented in future studies to inform this new scenario. It is also advisable to consult recent scientific literature and instruct local health authorities to provide specific information on horse flies as potential mechanical vectors in southern Chile and other locations, should they be considered a possible risk to public health.

These results are promising, and at the same time, provide a report of microorganisms with the potential to produce disease in animal and the human populations, paving the way for further studies on these and other emergent agents in insect vectors in Chile. Understanding the potential for mechanical or biological vector competence is critical to defining the precise role of *O. lata* in any pathogen cycling; the presence of DNA in an arthropod does not necessary imply that a transmission pathway exists, and one will need to assess its feeding process, pathogen surface viability, and the ability of the insect to acquire, disseminate and transmit each agent, via experimental infection studies.

## Author Contributions

**Conceptualization:** Christian R. González, Andrés Castillo, Jorge Fernández, Gillian Eastwood, Beatriz Cancino-Faure.

**Data curation:** Christian R. González, Andrés Castillo, Jorge Fernández.

**Formal analysis:** Christian R. González, Carolina Reyes, Andrés Castillo, Lara Valderrama, Lorena Llanos, Jorge Fernández.

**Funding acquisition:** Christian R. González.

**Investigation:** Christian R. González, Carolina Reyes, Lara Valderrama, Lorena Llanos.

**Methodology:** Christian R. González, Carolina Reyes, Andrés Castillo, Lara Valderrama, Lorena Llanos, Jorge Fernández, Beatriz Cancino-Faure.

**Project administration:** Christian R. González.

**Resources:** Andrés Castillo, Jorge Fernández, Gillian Eastwood.

**Software:** Andrés Castillo, Jorge Fernández.

**Supervision:** Christian R. González, Jorge Fernández.

**Validation:** Andrés Castillo, Jorge Fernández.

**Visualization:** Christian R. González, Andrés Castillo, Jorge Fernández.

**Writing – original draft:** Christian R. González, Andrés Castillo, Beatriz Cancino-Faure.

**Writing – review & editing:** Christian R. González, Andrés Castillo, Gillian Eastwood, Beatriz Cancino-Faure.

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
