## [Decision Letter · Decision Letter 0]

10 Apr 2024

Dear Dr. Eastwood,

Thank you very much for submitting your manuscript "Exploring the role of the black horsefly, Osca lata (Diptera: Tabanidae), as a mechanical vector: molecular evidence of pathogens in southern Chile" for consideration at PLOS Neglected Tropical Diseases. As with all papers reviewed by the journal, your manuscript was reviewed by members of the editorial board and by several independent reviewers. In light of the reviews (below this email), we would like to invite the resubmission of a significantly-revised version that takes into account the reviewers' comments. 

Expert reviewers in the research fields of vector biology, vector-borne diseases, and rickettsial pathogens have evaluated your manuscript and have considered that your study provides valuable preliminary insights into the prevalence of pathogens in the horse fly Osca lata in southern Chile. However, reviewers have also highlighted flaws in the methodology employed, mainly in the amplified and sequenced loci used to identify the rickettsia present in the samples. The PCR products amplified and sequenced appear too small and inadequate to characterize the rickettsia found at the species level, which is key to determining if the taxa found in these horse flies really belong to pathogenic rickettsia or not. Perhaps using Multilocus Sequence Typing (MLST) would be appropriate to robustly characterize the rickettsia isolates found in your samples. Moreover, the present study does not demonstrate the vector role of Osca lata. It would be necessary to nuance the manuscript more in light of the reviewers' comments. 

Finally, please ensure that all the data used in the present manuscript are publicly available, with a functional link to the repository in the manuscript.

The comments of the reviewers are enclosed for your consideration. I hope the information provided by the reviewers will be helpful for improving your analysis and the overall manuscript. All reviewers' comments should be addressed before the acceptance of the manuscript.

We cannot make any decision about publication until we have seen the revised manuscript and your response to the reviewers' comments. Your revised manuscript is also likely to be sent to reviewers for further evaluation.

Sincerely,

Winka Le Clec’h

Academic Editor

Nigel Beebe

Section Editor

Reviewer's Responses to Questions

**Key Review Criteria Required for Acceptance?**

**Methods**

-Are the objectives of the study clearly articulated with a clear testable hypothesis stated?

-Is the study design appropriate to address the stated objectives?

-Is the population clearly described and appropriate for the hypothesis being tested?

-Is the sample size sufficient to ensure adequate power to address the hypothesis being tested?

-Were correct statistical analysis used to support conclusions?

-Are there concerns about ethical or regulatory requirements being met?

Reviewer #1: (No Response)

Reviewer #2: The study's approach seems well-suited to its goals, using molecular methods to determine the prevalence of pathogens in horseflies from chosen areas in Chile. Picking Osca lata and these particular locations makes a lot of sense, especially given how much there still is to learn about this topic in Chile.

1) The aim/objectives could be improved by:

• Specifying species used 

• using terms like “prevalence of pathogens” 

• specifying pathogens 

• specifying molecular techniques used

2) Is the study design appropriate to address the stated objectives? Yes

3) The sample size was sufficient given the study being exploratory n=95. 

4) For clarity add a short sentence as to why nested PCR is used for Rickettsia detection but not for the other pathogens. e.g. increased specificity for Rickettsia detection

5) In Table 1, the nested PCR primers for Rickettsia were referenced as "This study." Please clarify if these primers were designed specifically for this study. 

5) Listing the original names of primers in Table 1 helps others repeat the study and understand where the primers came from. The gltA primers from Labruna et al. were originally named CS-78 etc. 

6) L157 is confusing as nested PCR is a specific type of two step PCR rather say conventional and nested PCR. 

7) Please add ethical or regulatory approvals that were obtained for this study

8) Were sequences obtained from this study submitted to GenBank? The manuscript makes no mention of this or accession numbers retrieved. Please submit to GenBank and add relevant accession numbers for validation of results.

Reviewer #3: Although the work demonstrates a correct methodology, they do not provide a sample size that is adequate to reach the conclusions they discuss. Having already performed the sequencing, they are not able to reach the species, having used specific primers for such an action. This gives uncertainty and concern in the possible results, since it is different to think of rickettsia rickettsii with r. parkeri, although both share genes, they behave differently, which is why the need to reach the species level for the PCR products that they amplified. So the recommendation would be to sequence with different genes and concatenate to reach genus and species.

**Results**

-Does the analysis presented match the analysis plan?

-Are the results clearly and completely presented?

-Are the figures (Tables, Images) of sufficient quality for clarity?

Reviewer #1: (No Response)

Reviewer #2: Here are a few suggestions:

1) Fig 1-4 are unclear and distorted, I can’t read phylogenetic tree labels. Please add higher quality images. the sample IDs in the figures "23-OTR-51," should explain what they mean and which sample site or species they’re from. Maybe add a SI table to show the information or a short sentence in the methodology section. 

2) Bootstrap values are missing from the trees. If they're missing because the support was not significant, it's helpful to mention this in your text to clarify the tree's interpretative value.

3) Subheadings need to be more detailed and precise. For eg. Subheading L145 should add “Sample preparation, DNA extraction and quantification”. 

Line 174-185 add sequencing details after PCR and then create a new subheading saying Phylogenetic analysis.

L197-203, It can be confusing to readers the way the percentages and totals are reported rather follow a consistent formatting across all instances where percentages and numbers are mentioned.eg. Among the 95 samples analyzed by PCR, pathogens were detected in 36.8% (n=35) of the samples. Specifically, Rickettsia spp. DNA was present in 29.4% (n=28) of the samples, Trypanosomatidae DNA in 5.2% (n=5)…. etc

L199-203 can be rewritten for clarity. Eg. In our study, out of the X number of horsefly samples analyzed, two samples (….%) exhibited co-infections with two distinct pathogens. Specifically, one specimen was co-infected with Filarioidea and Rickettsia, while another specimen showed co-infection with….

Reviewer #3: The results they show are alarming for public health, since they are focusing on zoonotic pathogens, however, it is necessary to have extracted the DNA from tissue with the ideal kit for tissue and not for blood. The images you show have poor quality, it is necessary to add images with better resolution more than 600 dpi. It is important to keep a table, which can compare which insects presented co-infections.

**Conclusions**

-Are the conclusions supported by the data presented?

-Are the limitations of analysis clearly described?

-Do the authors discuss how these data can be helpful to advance our understanding of the topic under study?

-Is public health relevance addressed?

Reviewer #1: (No Response)

Reviewer #2: The discussion/conclusion section does a great job of shedding light on the intriguing role of Osca lata in carrying pathogens in southern Chile. I have a few suggestions/comments for improvement : 

1) briefly comparing the pathogens in Osca lata with those in other local vectors like ticks or mosquitoes could give us even more insight into Osca lata's unique role in spreading diseases in the region.

2) Advocate for a "One Health" approach to discuss how understanding the role of Osca lata in pathogen transmission can inform integrated strategies to mitigate the risk of vector-borne diseases.

Reviewer #3: To reach these conclusions it is necessary to know the biology of insects in comparison to ticks or other potential vectors. Since the feeding process is different in each case. The use of words such as competent and potential is necessary, since demonstrating DNA does not mean that it is capable of transmitting it. It is important to add the points against this theory. There are different works that point to insects as potential vectors, however they mention their biological characteristics.

**Editorial and Data Presentation Modifications?**

Reviewer #1: (No Response)

Reviewer #2: (No Response)

Reviewer #3: (No Response)

**Summary and General Comments**

Reviewer #1: This manuscript by Gonzalez et al. reports on the molecular detection of several types of microbes in tabanid horse flies (Osca lata) collected from two regions in Chile. 95 specimens were dissected and mouthparts and guts were tested by PCR using primers that target Rickettsiae, Trypanosomatidae, and Filarioidea. The authors report detection of each of these taxa at appreciable prevalence, which comprises the first study of this type in horse flies from Chile and of this particular horse fly species in general. 

The manuscript is overall well-written, and the methods used are appropriate. However, it is my opinion that the manuscript requires some significant revisions before publication. In particular, some aspects of the introduction and discussion require more nuance/detail. Further, there are some issues surrounding the interpretation of the results with regards to the microbes detected being classified as “pathogens” from insufficient or contradictory evidence., and the broad use of the term “pathogen” inappropriately. 

Specific points are provided below. 

Line 1: I find the title a bit misleading as this manuscript does not directly assess the vector role of horse flies in any way. What is presented here is molecular surveillance/molecular detection of pathogens in horse flies. I would recommend to modify the title slightly by removing the reference to a vector role. Maybe something like “Molecular evidence of pathogens in the black horse fly, Osca lata, from southern Chile”, or something similar. 

Line 26: Please check the manuscript throughout for consistent use of the term “horse fly/horse flies”. As these are true flies, it should be two words, but in various parts of the manuscript it is written as one word incorrectly. 

Line 26-35: The authors make broad use of the term “pathogen” without evidence in the abstract and throughout the manuscript. While Dirofilaria immitis, which the authors detected, is a pathogen, the other sequences the authors detected do not appear to correspond to pathogens. Additional comments about this are given below, but the authors need to be more discerning when referring to D. immitis vs the other microbes that they detected throughout the manuscript and be clearer that the others do not appear to be pathogens. 

Introduction lines 88-102: I found the introduction with regards to the vector role of horse flies to be thin. In would be very useful and improve the manuscript if the authors expanded and gave more details about the types of pathogens that horse flies have been found to vector and the types of pathogens that have been detected in association with them in other regions. 

Similar to my comment above, information about Osca lata in particular is lacking in the introduction. What is the importance of this species more specifically? Why study it? What, if any, studies have been done in this species? Some of this is mentioned in the discussion but I think should also be in the introduction. 

Line 167: 390 bp product for Rickettsia is relatively small. Although the authors sequenced PCR products for further analysis, they themselves acknowledge that sequencing of this region is inadequate for species level identification. Therefore, related to my comment about the use of the term “pathogen” above, the authors should not refer to this is a pathogen. Indeed, from lines 205 of the results and lines 344 of the discussion, it appears that the sequence detected most closely matched the non-pathogenic Rickettsia, R. bellii.

Line 226: Similar to my point above, Crithidia dedva is an insect specific Trypanosomatid and not a pathogen. Please be more discerning when referring to this microbe and do not refer to it as a pathogen.

Reviewer #2: The study provides valuable preliminary insights into the prevalence of pathogens in Osca lata in southern Chile. Below are suggestions/comments based on abstract and introduction sections. I have included comments for the discussion under the conclusion section.

Abstract

4) "provides evidence of the molecular detection of parasites." , "parasites" is broad, specifically state which parasites. 

Introduction:

L113-115 should be removed from the intro as it states study results. 

L50-64 seems overdrawn and doesn’t directly relate to your study’s focus. Shorten it and directly introduce horseflies to the first paragraph.

In your introduction, several statements are left hanging without any references, for eg. L59-64, L78-82 etc.

Reviewer #3: It is necessary to amplify other genes to reach the species of each pathogen they indicate. based on the results obtained from sequencing. They should change or enrich the discussion of their work, since there are different species that can cause acute or subclinical diseases. Add the biology of insects describing the pros and cons of acting as a vector for certain pathogens. Use the terms potential and competent, differentiating which one your hypothesis falls into.

PLOS authors have the option to publish the peer review history of their article (what does this mean?). If published, this will include your full peer review and any attached files.

Reviewer #1: No

Reviewer #2: No

Reviewer #3: No
---

## [Decision Letter · Decision Letter 1]

21 Aug 2024

Dear Dr. Eastwood,

Thank you very much for submitting your manuscript "Molecular evidence of pathogens and endosymbionts in the black horse fly Osca lata (Diptera: Tabanidae) in Southern Chile" for consideration at PLOS Neglected Tropical Diseases. As with all papers reviewed by the journal, your manuscript was reviewed by members of the editorial board and by several independent reviewers. The reviewers appreciated the attention to an important topic. Based on the reviews, we are likely to accept this manuscript for publication, providing that you modify the manuscript according to the review recommendations. 

Sincerely,

Winka Le Clec’h

Academic Editor

Nigel Beebe

Section Editor

Dear Dr. Eastwood,

Thank you for submitting your revised manuscript to PLOS Neglected Tropical Diseases (PNTD-D-24-00263 - "Exploring the role of the black horsefly, Osca lata (Diptera: Tabanidae), as a mechanical vector: molecular evidence of pathogens in southern Chile").

We appreciate your efforts in amending your manuscript in response to the reviewers' remarks, critiques, and concerns. However, Reviewer 2 still has some minor critiques regarding the English language and sentence structure throughout the manuscript. I think these issues can be easily addressed and will benefit your manuscript.

The reviewers' comments are enclosed for your consideration. I hope their feedback will be helpful in improving the clarity and readability of your manuscript.

Thank you again for your interest in the PLOS Neglected Tropical Diseases journal.

Sincerely,

Winka Le Clec'h

Reviewer's Responses to Questions

**Key Review Criteria Required for Acceptance?**

**Methods**

-Are the objectives of the study clearly articulated with a clear testable hypothesis stated?

-Is the study design appropriate to address the stated objectives?

-Is the population clearly described and appropriate for the hypothesis being tested?

-Is the sample size sufficient to ensure adequate power to address the hypothesis being tested?

-Were correct statistical analysis used to support conclusions?

-Are there concerns about ethical or regulatory requirements being met?

Reviewer #1: No concerns

Reviewer #2: Mention that the 18S gene was amplified for Filarioidea detection. 

 I noticed areas where the sentences could be improved for clarity, L139-140, L145-147 and L179-

Also the PCR reaction components was excluded from the revised manuscript, please include this part.

Reviewer #3: It is a quite interesting study, which shows that horseflies can be mechanical vectors of some diseases. Both the general objective and the specific objectives are well defined. The methodology is correct, I would have liked a larger sample than the current one, however I think it is enough to start the study, which undoubtedly needs to continue.

**Results**

-Does the analysis presented match the analysis plan?

-Are the results clearly and completely presented?

-Are the figures (Tables, Images) of sufficient quality for clarity?

Reviewer #1: No concerns

Reviewer #2: L217-218 can be phrased better, rather say “Twenty-two specimens, collected in 2021 (n=8) and 2022 (n=14), tested positive for one or more of the detected microorganisms.”

Same with L192-193

L232- it would be more accurate to say PCR amplicons

In figure 2 caption “Chilean samples from this study are highlighted in red”, the highlighted portion is green ? same with figure 3 and 4. 

Also in this section there are sentences that could be written better for clarity like L215- instead of "In total, 95 field-caught female specimens" you could say " A total of 95 field-caught female specimens" and "Of these, 22 specimens (eight in 2021 and 14 in 2022) tested positive.." etc.

Ensure Table 1 includes all necessary lines and data for clarity.

Reviewer #3: The results are adequate, although it is possible to increase them by proposing the role of horseflies in the transmission, since we know that insects play an important role in the transmission of trypanosomes and filariae, this is not the case in the case of rickettsiae, therefore the The presence of these MOs in tabanos may be somewhat indicative, but may not imply a real threat to public or veterinary health.

**Conclusions**

-Are the conclusions supported by the data presented?

-Are the limitations of analysis clearly described?

-Do the authors discuss how these data can be helpful to advance our understanding of the topic under study?

-Is public health relevance addressed?

Reviewer #1: No concerns

Reviewer #2: sentences such as "This is the first study describing the molecular detection of microorganisms in horse flies in Chile, and in this species," and "Osca lata females avidly seek blood from different hosts, including humans, causing unquantified losses to tourism," could be rephrased for better readability.

Reviewer #3: It is important to point out the role of these insects in the transmission of diseases, and to differentiate them from the role of potential vector and competent vector, since these may be playing a role in remaining in nature and not in itself, a transmitting role, as in the case of rickettsia in particular.

**Editorial and Data Presentation Modifications?**

Reviewer #1: There are a few typographical & punctuation errors that should be corrected by careful editing, but I believe this can be done post acceptance.

Reviewer #2: (No Response)

Reviewer #3: Accept

**Summary and General Comments**

Reviewer #1: I have no remaining concerns regarding the scientific rigor and validity of the data and conclusions in this paper.

Reviewer #2: The comments from the initial review were addressed, however, the English language and sentence structure throughout the manuscript could be greatly improved for clarity and readability. Some sentences are verbose or awkwardly phrased, i mentioned a few instances above but there are many throughout the manuscript.

Reviewer #3: This project is quite important, and demonstrates one more actor in the complex of vector-transmitted diseases, it is necessary to develop greater precision when selecting the initiators, since they are very general, and it is necessary to carry out a second round now with initiators more specific, to reach the gender of the causing MO. It is also important to know what manifestations are or are not causing the reservoirs or if they are capable of delimiting the possible infection, and in turn, the possible immunological process.

PLOS authors have the option to publish the peer review history of their article (what does this mean?). If published, this will include your full peer review and any attached files.

Reviewer #1: No

Reviewer #2: No

Reviewer #3: No

Figure Files:

Data Requirements:

Reproducibility:

References

---

## [Editor Report · Decision Letter 2]

10 Sep 2024

Dear Dr. Eastwood,

We are pleased to inform you that your manuscript 'Molecular evidence of pathogens and endosymbionts in the black horse fly *Osca lata* (Diptera: Tabanidae) in Southern Chile' has been provisionally accepted for publication in PLOS Neglected Tropical Diseases.

Best regards,

Winka Le Clec’h

Academic Editor

Nigel Beebe

Section Editor

Dear Dr. Eastwood,

After receiving and reviewing your revised manuscript (PNTD-D-24-00263 - "Exploring the role of the black horsefly,* Osca lata* (Diptera: Tabanidae), as a mechanical vector: molecular evidence of pathogens in southern Chile"), I think it is now ready for publication in PNTD. Thank you for thoroughly addressing the reviewers' comments and critiques and revising the manuscript accordingly.

Best regards,

Winka Le Clec'h, PhD

---

## [Editor Report · Acceptance letter]

20 Sep 2024

Dear Dr. Eastwood,

We are delighted to inform you that your manuscript, "Molecular evidence of pathogens and endosymbionts in the black horse fly Osca lata (Diptera: Tabanidae) in Southern Chile," has been formally accepted for publication in PLOS Neglected Tropical Diseases.

Best regards,

Shaden Kamhawi

co-Editor-in-Chief

Paul Brindley

co-Editor-in-Chief
